# Psyllids in Natural Habitats as Alternative Resources for Key Natural Enemies of the Pear Psyllids (Hemiptera: Psylloidea)

**DOI:** 10.3390/insects15010037

**Published:** 2024-01-06

**Authors:** David R. Horton

**Affiliations:** Temperate Tree Fruit and Vegetable Research Unit, USDA-ARS, Wapato, WA 98951, USA; david.horton@usda.gov

**Keywords:** conservation biological control, hawthorn psyllids, willow psyllids, *Anthocoris*, *Deraeocoris*, *Trechnites*, *Prionomitus*

## Abstract

**Simple Summary:**

The pear psyllids comprise over 20 species of small, sap-feeding insects which include several of the most damaging pests in commercial pear orchards of Europe, temperate Asia, and the Americas. Efforts to control these pests historically have centered on the use of insecticides. Because psyllid outbreaks often may be prompted by insecticidal destruction of natural enemies, pear growers have begun to substitute biological control and conservation of natural enemies for some insecticide applications. Conservation of natural habitats to make alternative resources (prey and hosts) available to natural enemies of crop pests increasingly is being examined as a tactic for strengthening biological control in crops. This review shows that psyllids in natural habitats are important resources for species of predators and parasites which provide biocontrol of pear psyllids. Traits which affect suitability of non-pest psyllids to beneficials include how closely related they are to pear psyllids and type of seasonal life cycle, the latter shown to govern when (seasonally) non-pest psyllids in natural habitats are available to beneficials. Confirmation that natural habitats and these alternative resources lead to numerical increases in beneficials in orchards requires proof that key predators and parasites colonize orchards from natural habitats. Difficulties in obtaining this proof are discussed.

**Abstract:**

The pear psyllids (*Cacopsylla* spp.; Psylloidea) comprise ~24 species of sap-feeding insects distributed in Europe, temperate Asia, and (as introductions) in the Americas. These pear-specialized insects are among the most damaging and difficult to control pests in orchards. Biological control increasingly is being used to replace or partially replace insecticidal management of pear psyllids. Many key natural enemies of pear psyllids regularly occur in non-orchard habitats on native plants. The presence of beneficial species both in orchard and non-orchard habitats (here referred to as “spillover”) has prompted suggestions that native plants and their associated psyllids should be conserved as alternative resources for natural enemies of pear psyllids. The expectation is that the natural enemies will move from those habitats into psyllid-infested orchards. This review shows that psyllids in native habitats are important resources for several key predators and parasitoids of pear psyllids. These resources are critical enough that some beneficials exhibit an almost nomadic existence as they move between plant species, tracking the seasonal appearance and disappearance of psyllid species. In contrast, other natural enemies show minimal or no spillover between orchard and non-orchard habitats, which likely is evidence that they exhibit limited movement at best between orchard and non-orchard habitats. To show conclusively that spillover also indicates that a beneficial species disperses between native habitats and orchards requires difficult research on insect movement. This review concludes with a brief discussion of these difficulties and possible solutions.

## 1. Introduction

Over 20 million tons of pears were produced globally in 2020, dominated by Europe, the United States, and China [1]. The pear-feeding psyllids (Hemiptera: Psylloidea: Psyllidae: *Cacopsylla* spp.) are a group of 20+ species of small sap-feeding insects distributed in the pear-growing regions of Europe and Asia and as invasive introductions in North and South America [1]. These species develop only on species of pears (Rosaceae: *Pyrus*) and, in many growing regions, are the most damaging and difficult to manage insect pests of pears. The pear psyllids cause several types of damage, which lead to yield losses, downgrading of fruit, defoliation and reduced tree vigor, premature fruit drop, and vectoring of the Phytoplasma that causes pear decline [1]. The primary damage is marking of fruit by nymph-produced honeydew. Psyllid nymphs produce large quantities of liquid honeydew which drips onto fruit and is colonized by a sooty mold, and this leads to marking and downgrading of the harvested product. Orchards may become so sticky from honeydew that activities such as summer pruning, thinning of fruit, and harvest are hampered. Outbreaks of nymphs cause defoliation of the pear tree, reduced fruit size and quality, and premature fruit drop. If damage is extensive enough, yield losses may carry over between consecutive years.

Synthetic insecticides have been the foundation of psyllid management in pear orchards since the 1940s [1]. This emphasis began to be reassessed in the 1960s and 1970s due to the arrival of insecticide resistance and with a realization that outbreaks of psyllids often were caused by grower practices [2]. The pear psyllids are referred to as “induced” pests, in that outbreaks tend often to be associated with destruction of predators and parasites caused by use of broadly toxic insecticides [2,3,4]. This realization has prompted a slow shift into integrated programs in which selective insecticides, biological control, and cultural or horticultural practices are used as substitutes for high-toxicity psyllicides. A core tactic of these IPM approaches is to use practices that conserve natural enemies but which do not lead to economically damaging levels of psyllids [5,6,7,8,9]. Conservation practices include increased emphasis in monitoring of predators and parasitoids in orchards and in manipulating types, quantities, or timings of insecticides to minimize the effects of psyllicides on them [7,8,9].

An additional tactic of conservation biological control is to exploit the services of beneficial species whose populations spend some portion of the growing season outside of the crop [10,11,12]. Beneficial species feed upon or parasitize non-pest arthropods in non-crop habitats, either for maintenance or for reproductive increase. The expectation is that population increase by natural enemies in these habitats is followed by dispersal into the crop as resources in native habitats become depleted or because the crop becomes a source of prey and hosts [10,11]. There is an undeniable and compelling logic to this idea, and this has led to a large body of research where the theory has been examined [12,13]. The question in many of these studies is whether it makes biological and economic sense to conserve some level of plant and arthropod diversity in crop-growing regions. Published syntheses of these studies show that this can be a difficult question to answer [12,13,14].

The idea that non-orchard habitats might be important resources for natural enemies of the pear psyllids began to receive attention in the 1980s with suggestions that certain plant species should be conserved to encourage population increase by natural enemies of pear psyllids [15]. The climatic requirements of pear trees means that pear orchards occur in regions characterized by habitats and climates that are suitable for native deciduous trees and shrubs (Figure 1). Pears require an interval of low (but not severely cold) temperatures in winter to complete dormancy and to produce a vigorous tree in spring [16]. Trees also demand ready access to water, provided generally through irrigation [16]. Concentrations of pear orchards in western North America, for example, are in foothills and interior river valleys where mountain snows contribute to irrigation and where topography, rivers, and lakes lend protection against severe winter temperatures [16]. Consequently, orchards often abut landscapes of deciduous upland trees and shrubs, shrub-filled rangeland, and riparian areas of water-loving trees and shrubs (Figure 1). It is this natural diversity which, in part, led to early suggestions about habitat conservation and psyllid biocontrol [15,17]. Plant species listed for possible conservation included those which host native psyllids, as the assumption was that species of natural enemies which preferentially feed on or parasitize non-orchard psyllids are also likely to attack pear psyllids [15,17]. Indeed, psyllids in native habitats have long been known to be a species-rich resource for a large community of predators and parasitoids, several of which we now know are crucial natural enemies of pear psyllids [15,18,19,20,21,22,23,24,25,26,27,28,29].

Any efforts to augment biological control of pear psyllids through habitat conservation must first establish that native habitats and the prey or host resources of those habitats indeed are important to key natural enemies. In this review, I summarize a large and scattered literature to show that psyllids in habitats outside of the orchard ecosystem are significant resources for key natural enemies of the pear psyllids. I have focused efforts on a core set of predators and parasitoids identified as key natural enemies of pear psyllids. The survey shows that non-pest psyllids are important prey and hosts for this core set of natural enemies and that psyllid species closely related to the pear psyllids often are the targets of those natural enemies. I examined seasonal life cycles of important prey and hosts to determine whether psyllid variation in life cycles governs seasonal availability of different species to natural enemies. This analysis leads to the observation that the seasonal appearance and subsequent disappearance of different psyllid species cause an almost nomadic existence for predators and parasitoids as they track their prey or hosts through the non-crop landscape. In particular, several species of willow- and hawthorn-associated psyllids are essential early-season prey and hosts of natural enemies. This synthesis concludes with a brief discussion of the concept of “spillover”, used here to describe the regular presence of a beneficial species both in native habitat and in pear orchards. Differences among key natural enemies in spillover are used to illustrate that the effects of habitat diversification on biocontrol services are likely to be unequal among species of natural enemies. Those predators or parasites which specialize on pear psyllids or which specialize on native psyllids (i.e., exhibit low “spillover”) are unlikely to see their impact on pear psyllids alter in any meaningful way with habitat diversification. In the end, however, research on insect movement is required if we are to prove that spillover does indeed mean that the beneficial disperses between crop and non-crop habitats. The challenges which accompany this type of research are discussed briefly.

## 2. Sources of Information Used in Synthesis

Taxonomy of the pear psyllids follows Cho et al. 2017 [30]. Currently, 24 species of *Cacopsylla* are known to develop on pear [30]. The majority (16) of these species occur in the eastern Palearctic region [30]. Virtually all that is known about biological control of the pear psyllids is limited to species in the western Palearctic region and North America [1]. Because of this shortcoming, the current review is limited to the growing regions of Europe and western North America. I additionally have limited this synthesis to a core set of six natural enemies, consisting of four predatory true bugs in the Anthocoridae and Miridae (Hemiptera) and two parasitoids in the Encyrtdiae (Hymenoptera). The pear psyllids are attacked by a large assemblage of parasites and predators. By limiting this review to the set of natural enemies which I have judged to be of key importance in biological control of pear psyllids, the synthesis becomes manageable. My justifications for including a species in the list are shown below. Records showing that psyllids other than pear psyllids are important alternative prey or hosts for key natural enemies were accumulated from literature accounts of predator biology and feeding preferences, web-based or literature records of parasitism, and from personal observations and collecting. Patterns in seasonal presence of different psyllid species were determined from published syntheses of psyllid life cycles (reviewed in [31]) or from personal observations.

## 3. A Brief Introduction to the Psylloidea

Psyllids (Hemiptera: Psylloidea) are small sap-feeding insects related to aphids, scale insects, and whiteflies (Figure 2). Global diversity of the Psylloidea exceeds 4000 known species, with an unknown number of species awaiting description [32,33]. Psyllids occur in all biogeographic regions except Antarctica [32]. Seven families currently are recognized [32]. Psyllidae and Triozidae—both with over 1000 described species—are the most species-rich of the seven families. All of the pear psyllids are placed in the genus *Cacopsylla* and the species-rich family Psyllidae. The genus *Cacopsylla* has primarily a north temperate distribution, with hosts often in Rosaceae, Salicaceae, and Rhamnacae [32,34]. *Cacopsylla* proves to be an important alternative resource for key predators and parasitoids of the pear psyllids, as is elucidated in this review.

Life stages of psyllids consist of the adult insect, egg, and an immature form (nymph) having five developmental stages or instars (Figure 3). The nymphal stage is the primary target of predators and most parasitoids, although the egg stage also can be an important target of predators. Most species of psyllids have a tendency towards monophagy, with many species limited to development on a single plant genus or plant species [35,36]. This tendency toward monophagy has important consequences for conservation biological control, in that psyllid species which are alternative hosts or prey of parasites and predators in natural habitats will not themselves be found reproducing in pear orchards, causing damage. Finally, the psyllids exhibit a bewildering array of seasonal life cycles largely governed by climate, host plant, and phylogeny [31]. These different strategies allow psyllids to survive and flourish even in extreme climates or in regions where the host plant fluctuates in seasonal availability or quality [31]. Psyllid species differ in annual numbers of generations, presence or absence of a physiological diapause, and overwintering stage [31]. One important consideration for the purposes of this review is that type of life cycle often governs when (seasonally) prey or hosts are available to predators and parasites. The effect of psyllid life cycle on resource use by natural enemies is to be examined in detail below.

## 4. Six Key Natural Enemies of the Pear Psyllids

Six species of natural enemies are targeted for review (Figure 4 and Figure 5): four species of predatory true bugs (Hemiptera: Anthocoridae, Miridae) and two species of parasitoids (Hymenoptera: Encyrtidae). This focus is prompted by three factors. First, limiting this review to a core group of six species makes the review manageable. Second, all six species contribute extensively to biological control of pear psyllids. Indeed, the trio of predators and parasites in Europe selected here has been described to comprise the only species within the large complex of natural enemies in orchards which are “closely tied” to the pear psyllids [24] and consistently effective in reducing their numbers. Emphasis on these species is not meant to imply that other taxa are unimportant in orchards. Opportunistic predators such as spiders (Araneae), earwigs (Dermaptera), and certain Miridae (e.g., *Campylomma verbasci*), as well as predators which generally prefer non-psyllid prey, often contribute to biological control of pear psyllids [7,9,24,37]. However, these other predators are much less closely tied to psyllids than the targets of this review. Third, as will become apparent, each key species exhibits a modest to strong (sometimes obligatory) preference for psyllids, preferences which extend often to psyllids outside of the orchard and on native trees and shrubs. Since one intention of this review is to suggest that the presence of non-pest psyllids in habitats near orchards contributes to biological control of pear psyllids, a necessary stipulation is that there is some level of association between key beneficials and psyllids in general.

Key true bug predators include three species of minute pirate bugs (or flowerbugs) in the genus *Anthocoris* and a predatory mirid, *Deraeocoris brevis* (Uhler) (Figure 4). The bugs prey on the nymph and egg stages of pear psyllids. Both adults and nymphs are predaceous (Figure 4). All four species are multivoltine and consequently may be present in pear orchards at any time of the season, including in winter as overwintering adults [38]. The key species in European pear orchards is *Anthocoris nemoralis* (Fabricius) (Figure 4A), which is distributed throughout Western Europe and the Mediterranean Basin and eastwards at least to the Black Sea [39]. *Anthocoris nemoralis* has been stated in multiple accounts to be the most common and important predator of pear psyllids across an impressively large growing region: Great Britain [4], Spain [40], France [15,24,41], Israel [42], Turkey [43], Italy [44,45], Greece [46], Serbia [47], and The Netherlands [48,49]. This bug has been described as “a permanent component” [24] of the natural enemy community in pear orchards throughout Europe and one—as is shown later in this review—which exhibits a close connection in general to psyllids. The role of *A. nemoralis* in North American orchards is filled largely by three other true bug species, the flowerbugs *Anthocoris antevolens* White and *Anthocoris tomentosus* Péricart and the mirid, *Deraeocoris brevis* (Figure 4B–E). These three species are visible elements of the predatory faunas in pear orchards of British Columbia, Washington, Oregon, and parts of California, although not all species occur in all regions. The three predatory bugs may co-occur in orchards of British Columbia and central Washington State [25,29,50,51,52,53,54,55,56,57]. Orchards in California seem mostly to host *A. antevolens* [58,59,60], while *D. brevis* (sometimes co-occurring with *A. antevolens*) is a source of psyllid biocontrol in pear-growing regions of southern Oregon [61,62], northern Oregon [29,63], and northcentral Washington [5,7,9].

Multiple lines of evidence support the selection of these four predatory bugs as key sources of psyllid biocontrol, including clear statements in the literature that the presence of these species or a subset of these species is needed to achieve biological control [4,15,24,41,45,46,47,48,49,50,51,52,53,54,55,56,59]. All four predators track infestations of pear psyllids [40,42,43,48,50,52,54,64]. These predators colonize orchards in response to an emerging psyllid infestation and build to high numbers, followed often by collapse of the psyllid infestation and dispersal. Field releases of *A. nemoralis* and *A. tomentosus* into pear trees have been shown to control pear psyllids [65,66]. The four mentioned predators are efficient in locating pear psyllids even at low densities of the pest. Molecular gut contents analysis showed that nearly 100% of *Anthocoris* spp. or *D. brevis* collected at multiple intervals from a *C. pyricola*-infested pear orchard were found to harbor the pear psyllid signal [57]. Even as densities of psyllids declined to very low levels (one to two psyllid nymphs per 10 leaves), between 30% and 40% of the predator specimens routinely carried the signal.

The other natural enemies to receive attention here are two species of parasitoids (Figure 5): *Trechnites insidiosus* (Crawford) and *Prionomitus mitratus* (Dalman) (Hymenoptera: Encyrtidae). Both species attack only psyllids, and (as will be shown) both have a preference for psyllids in the family Psyllidae and genus *Cacopsylla*. Both species are widespread in Europe, although possibly not quite as widespread or consistently present in pear orchards as *A. nemoralis*. Each was introduced intentionally into North America on multiple occasions beginning in the 1960s in efforts to control the pear psyllid *C. pyricola* [67]. Both species in fact were present in North America before release. *Trechnites insidiosus* likely arrived in North America in the early 1900s, hitchhiking on pear trees imported from Europe [67,68], while *P. mitratus* apparently is naturally Holarctic [69]. The encyrtids parasitize nymphal psyllids, inserting their eggs into mid-instar or later-instar nymphs [70,71,72] and emerging (as an adult wasp) from the mummified last-instar nymph (Figure 5C,D). Each species is multivoltine. *Prionomitus mitratus* overwinters in the adult stage [73], while *T. insidiosus* overwinters as a mature larva in the psyllid mummy [70]. Both *T. insidiosus* and *P. mitratus* are sources of biological control in Europe [15,24,41,43,74,75,76]. Parasitism rates for *T. insidiosus* in Europe have been shown to exceed 25–50% in some orchards (reviewed in [37]), while late-season parasitism of *C. pyri* by *P. mitratus* may sometimes exceed 70% [15]. In North America, *T. insidiosus* increasingly is targeted for conservation in orchards [6,7,9]. Parasitism rates due to *T. insidiosus* can be higher than 50% in reduced-insecticide orchards of Washington, Oregon, and California (reviewed in [37,67]). Adults of *T. insidiosus* are among the most common natural enemies (predatory or parasitic) collected on beating sheets or sticky traps in some orchards [77,78,79]. The parasitoid *P. mitratus* in North America—as in Europe—is a common parasite of native psyllids [29,69,80]. However, in contrast to what is observed in Europe, *P. mitratus* in North America seems not to be an important parasite of pear psyllids. This dichotomy in the behavior of European and North American populations of *P. mitratus* is examined below.

Traits which separate the genus *Anthocoris* from related genera are summarized in monograph treatments of the European [39] and the North American [81] faunas of Anthcoridae. These publications also include keys with which to separate *A. nemoralis* from other *Anthocoris* and to separate *A. tomentosus* and *A. antevolens* from other North American species. A key with which to identify *Deraeocoris brevis* is available in a monograph of the North American species of *Deraerocoris* [82]. Taxonomic treatments of the two encyrtid parasitoids are also available. *Trechnites insidiosus* is readily separated from other European species by a key included in a 2009 overview of European *Trechnites* [83]. Delvare et al. (1981) provide a thorough and heavily illustrated description of *Prionomitus mitratus* which can be used to separate this species from related Encyrtidae [84].

## 5. Psyllids in Natural Habitats as Alternative Resources for Key Natural Enemies

### 5.1. Predators

Key predators of pear psyllids routinely prey on psyllid species outside of orchards. Evidence for this behavior first arrived in the late 1950s and early 1960s with two now-classic field studies of *Anthocoris nemoralis*. The first of these studies followed the predatory activities of *A*. *nemoralis* in woodland habitats of southern England and showed that overwintered *A*. *nemoralis* almost exclusively colonized three species of psyllid-infested shrubs while ignoring species of shrubs within the study area which are not hosts of psyllids [19]. Later generations shifted to other plant taxa as the psyllids disappeared. The second study was conducted during the early 1960s in stands of psyllid- and aphid-infested broom (*Cytisus scoparius*; Fabaceae). A serum test was used to identify prey remains in field-collected *A. nemoralis* [85]. Results of these assays showed that *A*. *nemoralis* selected psyllid prey over aphid prey, while a congeneric species—the generalist *Anthocoris nemorum* L.—showed no selectivity [85]. In North America, the late-1930s arrival of the pear psyllid *C. pyricola* in Washington State and its subsequent spread in the following decades prompted the initial research on North American predators of the pear psyllids and provided the first evidence that *A. antevolens*, *A. tomentosus*, and *D. brevis* feed on *C. pyricola* [50,53,54,59]. Observations that these species also are found outside of orchards in association with non-pest psyllids began to arrive at about the same time [18,20,60].

A list of psyllid species preyed upon by the core predators *Anthocoris nemoralis*, *A*. *tomentosus*, *A. antevolens*, and *Deraeocoris brevis* are shown in Table 1. For completeness, I include records and observations for the pear psyllids to accompany the larger set of records from natural habitats. Evidence that key predators of pear psyllids prey on non-pest psyllids falls into two categories differing in strength of support (Table 1). “Direct evidence” includes field observations of feeding, numerical increases by predators in response to psyllid infestations (combined often with declines in psyllid numbers), observations of immature predators intermixed with infestations of psyllids, and laboratory-based feeding trials (Table 1: Direct evidence). Evidence of lesser strength (Table 1: Other evidence) consists of statements that the predator was observed on psyllid-infested plants or on plant species known to host psyllids. Records in which the population of psyllid nymphs consisted of multiple species are listed as “assemblages of species” rather than as a single identified psyllid species, given uncertainties in knowing which psyllid species within the assemblage were targets of the natural enemy.

Field observations and feeding trials have identified more than two dozen species of psyllids as prey of *A. nemoralis*, while an additional 20 species of psyllids are listed from accounts of *A. nemoralis* being present on psyllid-infested plants or on plant species which host psyllids (Table 1: *Anthocoris nemoralis*). Many “Direct evidence” records are for insects in the family Psyllidae (14 of 27 records; 52%) and genus *Cacopsylla* (11 of 27 records; 41%), as are many of the remaining (“Other evidence”) records (Table 1). In the combined evidence, six families of psyllids in total are listed. The only family not represented is the Mastigimatidae. Certain host plant assemblages are particularly visible in the records—five species of pear psyllids (all *Cacopsylla*), four species of hawthorn psyllids (*Cacopsylla*), six species of ash psyllids (*Psyllopsis*), and an unknown number of willow psyllids (*Cacopsylla* spp.). Five of the six *Psyllopsis* records are from laboratory-based feeding trials rather than field observations. Several species of psyllids from outside of the native range of *A. nemoralis* also appear as records (Table 1: asterisks) and consist of invasive introductions from Australia, Asia, and South America generally in association with introduced ornamental plants. Four of these records (*A. uncatoides*, *T. adventicia, C. schini*, and *G. brimblecombei*) are from coastal regions of California, where *A. anthocoris* has become established on psyllid-infested ornamental species [88,91]. These records show that *A. nemoralis* is willing to feed on psyllid species with which it has no co-evolutionary history. Plant hosts of psyllid species in Table 1 consist almost exclusively of trees and woody shrubs.

In North America, accounts that *A. antevolens* and *A. tomentosus* are found outside of orchards associated with psyllids appeared initially in the 1960s with records of the insects on *Cacopsylla*-infested willows [18,60]. Later observations showed that both anthocorids occur disproportionately on tree and shrub species which are hosts to psyllids [26] and that both regularly occur on these plants together with psyllid prey [20,25,29]. While the mirid *Deraeocoris brevis* is more of a generalist than *A. tomentosus* and *A. antevolens,* it also regularly occurs on plants which host psyllids [25,29,82]. Direct evidence that *A. tomentosus*, *A. antevolens*, and *D. brevis* feed and develop on psyllids is less available than that for *A. nemoralis* (Table 1: Direct evidence). All three species readily consume the pear psyllid *C. pyricola* in laboratory trials, and all are found as adults and nymphs in psyllid-infested orchards. Adults and nymphs of the three species also are found together with nymphs of *Cacopsylla* on *Salix* (Table 1: Direct evidence). The *Salix* psyllids listed here include a taxonomically difficult group of univoltine, catkin-specialized species found on multiple species of willow, and a multivoltine psyllid (*Cacopsylla alba*) apparently limited to catkins and foliage of a single species, the sandbar willow *Salix exigua* (Figure 2E,F). Other evidence includes observations of predators on psyllid-infested plants or on plant taxa known to host psyllids (Table 1: Other evidence). I have collected *D. brevis* from psyllid-infested willows, antelope bitterbrush, Scotch broom, and pear, often together with *Anthocoris* [29]. *Deraeocoris brevis* also is found on plants which host psyllids in the family Triozidae and which do not seem to be regular sources of *A. antevolens* or *A. tomentosus*, such as stinging nettle (*Urtica gracilis*; Urticaceae) and weedy species of *Solanum* or *Lycium* (Table 1: Other evidence). Records for the North American species include plant species which in North America do not host any native psyllids, including pear, Scotch broom, and European hawthorn (Table 1: asterisks). The latter host plant was found to harbor *Anthocoris antevolens* together with the introduced hawthorn psyllid, *Cacopsylla peregrina* [29].

### 5.2. Parasitoids

Both parasitoid species almost exclusively parasitize psyllids from the family Psyllidae (Table 2). As in Table 1, records for pear psyllids are included here for completeness. The earliest host records for both species were of pear psyllids: *Trechnites insidiosus* in 1909 from *C. pyricola* [68]; and *Prionomitus mitratus* in the 1920s from the univoltine pear psyllid *C. pyrisuga* [99]. The initial *T. insidiosus* record surprisingly is from North America rather than from the parasite’s native Old World region, which shows that this species was present in North America well before it was introduced into the western US in the 1960s. *Trechnites insidiosus* specializes on pear psyllids [83], with records from four species (Table 2: *Trechnites insidiosus*). Other than the pear psyllids, only three other species of psyllids (all *Cacopsylla*) have been reported to host *T. insidiosus* (Table 2). European records include the apparently rare use of a hawthorn psyllid [24] and a mountain ash (*Sorbus*) psyllid [28]. The hawthorn account consists of a few records in one year at a single location in southern France [24], while the mountain ash records consist of two specimens from several years of collecting in Serbia [28]. The non-pear record from North America is for a native *Cacopsylla* (*Cacopsylla alba*) which develops on a species of *Salix* (Table 2). Parasitism of *Cacopsylla alba* by *T. insidiosus* is relatively common (Figure 6), although it should be noted that there is some uncertainty at this time in whether the willow- and pear-associated populations of *T. insidiosus* are a single intermixing population. Thus, while the willow-associated parasite keys to *T. insidiosus*, differences in sex ratio and possibly in phenology of the pear and willow populations of the parasite may mean that the two populations represent host-specialized biotypes or (potentially) separate species [29].

The second parasitoid of interest, *Prionomitus mitratus*—while also limited to psyllid hosts—is much more of a generalist than *T. insidiosus* (Table 2: *Prionomitus mitratus*). Hosts of this Holarctic species comprise multiple species and several genera of psyllids. The combined Old and New World records total more than 30 species of psyllids in 10 genera (Table 2). All but two records (from the families Aphalaridae and Carsidaridae) are from Psyllidae. All known hosts of *T. insidiosus* and *P. mitratus* develop on trees or shrubs (Table 2).

## 6. Psyllid Life Cycles Force a “Nomadic” Existence by Key Natural Enemies

### 6.1. Common Life Cycles of Cacopsylla Psyllids

The close connection between natural enemy species and psyllid prey or hosts has implications in the ecology of these beneficials, particularly in movements by natural enemies through the landscape. The seasonal life cycle of psyllids substantially determines what species of psyllids may be available to natural enemies at a given time of year. Constraints imposed by life cycle mean that eggs and nymphs (i.e., the life stages targeted by beneficials) of many psyllid species are present only for a short interval each year. A consequence of this seasonality is that natural enemies of psyllids may exhibit an almost “nomadic” [21,42] existence as they track the arrival and disappearance of different psyllid species throughout the season and across the host plant landscape. As psyllid species disappear from the landscape due to life cycle, “nomadic” predators and parasitoids of those psyllids are forced to search for new resources. One possibility, of course, is that these new resources could include psyllid-infested pear orchards.

The genus *Cacopsylla*—shown above to be a target of key natural enemies—provides a useful illustration of how life cycle may differ among even closely related species (Figure 7). Several examples showing how *Cacopsylla* life cycles impose constraints on natural enemies of those psyllid species are shown in the following subsection. *Cacopsylla* species almost invariably exhibit one of three types of life cycle: a multiple-generation cycle in which the adult insect typically is of the overwintering stage (Figure 7A) or one of two single generation cycles (Figure 7B,C). Virtually all pear psyllids have a multivoltine life cycle [30]. Activity begins in late winter or early spring with the emergence of adults from wintering quarters in shelter plants and pear trees (Figure 7A). This is followed rapidly by mating and egg laying (Figure 7A: presence of eggs shown as yellow ovals). Hatching of eggs is followed by nymphal development (Figure 7A: blue bars show presence of nymphs) and arrival of the first-generation adults. Production of subsequent generations ensures that eggs and nymphs are present for much of the season (Figure 7A). Other *Cacopsylla* having a multivoltine life cycle appear to be limited mostly to a few *Salix*-associated species and to several species on evergreen shrubs [29,31]. One multivoltine species in North American is the willow psyllid *Cacopsylla alba* (Figure 2E) shown earlier (Table 1) to be an important resource for key predators of *C. pyricola*. This willow specialist is included below in the discussion of “nomadism”.

Virtually all other *Cacopsylla* apparently are univoltine [31]. Species having this life cycle overwinter as adults off of the host plant (often on conifers) or as eggs inserted into the wood of the host (Figure 7B,C). Both life cycles substantially shrink the interval over which nymphs are present (Figure 7B,C: blue bars). *Cacopsylla* having this life cycle are among the first psyllids to be active in late winter and include species that are early-season resources for natural enemies of pear psyllids. Species which overwinter as adults return to the host in late winter or early spring for egg laying (Figure 7B). The spring generation of eggs and nymphs produces new adults which disperse in late spring or early summer from the host plant onto shelter plants where they eventually overwinter (Figure 7B). Consequently, the host plant may be free of psyllids for much of the year, with eggs and nymphs available to natural enemies only for a brief interval in spring and early summer (Figure 7B). This life cycle is common for species which develop in catkins of *Salix* [31], consisting of taxonomically difficult assemblages of *Cacopsylla* species both in Europe [101,102] and North America [103,104]. Other *Cacopsylla* having this life cycle include several species on hawthorn (*Crataegus*)—shown below to play a prominent role in predator and parasite nomadism (see following subsection)—as well as species on *Prunus*, *Malus*, *Sorbus*, *Rhamnus*, and (rarely) *Pyrus* [31]. A second type of univoltine life cycle is shown by species which overwinter in the egg stage (Figure 7C). Eggs are deposited into the wood of the host in autumn. Overwintered eggs hatch in late winter or early spring, followed by a single generation of nymphs. New adults appear in early summer and enter a period of reproductive quiescence on or off of the host plant, which ends in autumn as eggs destined for wintering are deposited (Figure 7C). Nymphs are present only in spring and early summer (Figure 7C). *Cacopsylla* having this life cycle again include a species on *Crataegus* (*Cacopsylla peregrina*; Figure 7C) as well as species on *Malus*, *Sorbus*, or *Ulmus* [31].

### 6.2. “Nomadic” Activities of Predators and Parasites of the Pear Psyllids

The predatory bugs *Anthocoris nemoralis* (in Europe), *A. tomentosus* and *A. antevolens* (in North America), and the parasitoid *Prionomitus mitratus* (in Europe) are used to illustrate how life cycles of psyllids lead to seasonal tracking of prey or hosts throughout the landscape by natural enemies of the pear psyllids. Each example also illustrates the value of psyllid species which are active very early in the season, especially univoltine hawthorn and willow psyllids. The first study was conducted in 1959/60 by N.H. Anderson [19], who monitored the seasonal movement of *A*. *nemoralis* populations through a woodland landscape in southern England. Anderson showed that the predator preferentially colonized host plants of psyllids early in the season before dispersing in later generations onto other plant taxa once the univoltine psyllids had disappeared (Figure 8). Adults emerged from overwintering sites in March and colonized tree and shrub hosts of psyllids, namely willows, hawthorn, and Scotch broom, while being virtually absent from species which do not host psyllids, such as lime, oak, beech, and other trees (Figure 8: Overwintered adults). Willow was colonized earlier than hawthorn and broom, beginning with the arrival of the predator on psyllid-infested catkins [19]. Psyllids included univoltine species of *Cacopsylla* on willow and hawthorn and two species (*Arytainilla spartiophila* (univoltine) and *Arytaina genistae* (bivoltine)) on broom. Egg laying by *A. nemoralis* began in April. The first nymphal generation was completed almost entirely on psyllid-infested hosts (Figure 8: First-generation *Anthocoris* nymphs). First-generation adults began to appear in late May, followed by dispersal onto summer hosts as psyllids on willow and hawthorn disappeared and aphids began to appear on summer hosts (Figure 8: First-generation adults). Thus, few first-generation adults remained with the plants upon which they had developed. Only a small second generation of *A. nemoralis* nymphs was produced on summer hosts (Figure 8: Second-generation *Anthocoris* nymphs), even though adults of the first generation were abundant. Consequently, the second generation of adults also was small (Figure 8). Many first-generation adults apparently failed to reproduce but instead entered reproductive diapause, possibly in response to the absence of psyllid prey [19].

A second example follows seasonal resource use by *Anthocoris tomentosus* and *A. antevolens* in a woodland/riparian habitat in central Washington State [25,29]. Both predators were monitored on several willow (*Salix*) species, *Quercus* (oak), and three species of *Populus* (Figure 9). While psyllid-infested hawthorn and willow catkins together are early-season resources in Europe for *A. nemoralis*, this early-season role in North America—which has no native hawthorn psyllids—is filled for *A. antevolens* and *A. tomentosus* by various willow species [18,20,25,29,60]. Willows in the study area are hosts to an assemblage of catkin-specialized psyllids in the genus *Cacopsylla* (Figure 10A–C). These univoltine species are active very early in the season and available to natural enemies only very briefly (Figure 9: *Salix* spp. (horizontal yellow bar)). Psyllids become active in late winter as they return to *Salix* from overwintering sites in conifers, well before catkins are available (Figure 10A,B). As bud scales spread, females oviposit into swelling catkins (Figure 10B). Densities of nymphs in catkins can be impressively high (Figure 10E,F), to the extent that infestation may result in the destruction of catkins [29,105]. Overwintered *A. antevolens* and *A. tomentosus* begin to appear in psyllid-infested catkins by early- to mid-April, followed by egg laying and the arrival of early-instar nymphs (Figure 9: *Salix* spp. (horizontal red line)). Late-instar nymphs of predators may be abundant by mid- to late-May in mature catkins, just as new adults of *Cacopsylla* are emerging and dispersing from the *Salix* host (Figure 9: *Salix* spp.). As the psyllids disappear, the adult anthocorids either disperse to other plant species or—if the willow stand becomes infested by aphids (*Chaitophorus* sp.)—remain on the *Salix* host into late summer (Figure 9: *Salix* spp. (horizontal red lines)).

A second spring resource for *A. antevolens* and *A. tomentosus* in the central Washington study area is the multivoltine psyllid *Cacopsylla alba* on the willow *Salix exigua* (Figure 9: *Salix exigua* (horizontal yellow bar)). Unlike most other willow species, *S. exigua* produces catkins over the duration of the growing season [106] which presumably has contributed to the multivoltine life cycle of *C. alba*; these catkins are attractive to egg-laying *C. alba* (Figure 2F). However, while catkins of *S. exigua* are available into late summer, they are produced sporadically and somewhat irregularly across different stands of the host plant. This trait leads to infestations of *C. alba* that are quite patchy both spatially and temporally [29]. *Anthocoris tomentosus* and *A. antevolens* are present on *S. exigua* from spring well into late summer, albeit often with an irregular distribution as they collect at sites or in stands infested by *C. alba* (Figure 9: *Salix exigua* (horizontal red line)). Plant taxa colonized later than the willows included *Populus* and *Quercus* (Figure 9: *Quercus garryana* and *Populus* spp.). These species do not host psyllids and consequently are free of *Anthocoris* in early spring but, by late spring and early summer, are infested by aphids. Aphid infestation leads to the colonization of these tree species by dispersing *Anthocoris* and the presence of adults and nymphs of the predators into late summer (Figure 9: *Quercus* and *Populus* (horizontal red lines)).

Two examples illustrate how the parasitoid *Prionomitus mitratus* is forced to shift between host species because of host life cycle. Jerinić-Prodanović et al., in 2010 and 2019 [28,107], collected psyllid mummies between April and November over a 14-year period from 51 locations in Serbia, at elevations ranging between 75 and 1150 m above sea level. The parasitized nymphs were identified as species and then set aside to allow the emergence of parasitoids. Eighty of the eighty-two nymphs found to have been parasitized by *P. mitratus* were species of *Cacopsylla* (Figure 11). Hosts included psyllids which develop on hawthorn (three species of *Cacopsylla*), apple, buckthorn (*Rhamnus*), prune, and pear (four species). Seven of the species were univoltine (Figure 11). All but one of the univoltine psyllids overwinter as adult insects (Figure 11). The univoltine species were attacked only in April through June (Figure 11), as nymphs would have completed development and been gone by early- to mid-summer. Species acting as hosts for *P. mitratus* later than mid-summer consisted entirely of multivoltine pear psyllids (Figure 11).

A second study with *P. mitratus* found that univoltine psyllids are used by the parasite to fill a seasonal gap in in which the pest psyllid (the pear psyllid *C. pyri*) is not available. Nguyen and Delvare 1982 [23] untangled the confusing sequence of psyllid hosts used in early spring by *P. mitratus* in southern France (Figure 12; modified from Figure 2 in Nguyen and Delvare 1982 [23]; a version of this figure also has been shown elsewhere [1]). Emergence of wasps from overwintering quarters is not synchronized with early-season phenology of the multivoltine *C. pyri*. However, wasp phenology does match the phenology of univoltine *Cacopsylla* on hawthorn or on pear (Figure 12). Wasps emerge in late winter or early spring and parasitize nymphs of the hawthorn psyllid *C. melanoneura* and the pear psyllid *C. pyrisuga* (Figure 12). Parasitism by subsequent generations of *P. mitratus* (Figure 12: G1 and G2) shifts to a second univoltine hawthorn psyllid, *C. crataegi*, and to the multivoltine pear psyllid *C. pyri* (Figure 12). *Prionomitus mitratus* shifts entirely to parasitizing *C. pyri* as the univoltine psyllids have disappeared for the year. Those univoltine species which act as hosts for the wasps preceding the summer transfer to *C. pyri* were referred to as “relay hosts” by Nguyen and Delvare 1982 [23].

## 7. Linkage between Orchard and Non-Orchard Habitats Varies with Natural Enemy

### 7.1. Spillover

The previous sections confirm that psyllid species in habitats outside of the pear orchard are alternative resources for key natural enemies of the pear psyllids. With that confirmation, the logical next step is to discuss whether those natural enemies also disperse into orchards and provide some level of biological control. A large body of literature has explored whether habitat diversity translates into biological control in crops or orchards. What often emerges from this research is that it can be frustratingly difficult to estimate the precise roles of non-crop resources in shaping levels of biological control in crops [12,13,14]. Because population processes outside of the crop are assumed to affect types or numbers of natural enemies in crop fields, biological control must be addressed at a landscape scale rather than at the scale of the individual field or orchard [11,108]. Analysis at this scale may create such a tangle of interactions—linking the targeted pests, beneficial species, non-pest prey or hosts, and their respective crop and non-crop habitats—that it is difficult to isolate and evaluate separate effects.

The earliest suggestion that habitat diversity should be conserved to strengthen biological control of pear psyllids dates to the early 1980s in southern France [15,22]. Trees and shrubs of several families were suggested as possible targets for conservation or planting. The plants which made this list were selected because they host psyllids as well as key natural enemies of pear psyllids [15]. The overviews from southern France create an impressively tangled web of interactions involving specialist and generalist predators (both insect and mite), several parasitoids and their hyperparasites, and multiple host plants of psyllids [15,24,41]. In this synthesis, I have instead focused on a much reduced set of species and interactions. The beneficial species chosen for review were selected because they have two important traits: they are primary sources of psyllid biological control in pear orchards; and they extensively utilize psyllids as prey or hosts. By simplifying the analysis to a few natural enemies and to a single alternative resource (psyllids), the expectation is that it becomes easier to examine whether habitat complexity and biological control of pear psyllids potentially are linked, at least for these key natural enemies.

An idealized representation of the linkage between native plants, their psyllids, and biological control in orchards by key natural enemies is shown in Figure 13 for two growing regions: southern France and Washington/Oregon in North America. The figures illustrate a concept referred to as “spillover” [11], which is used here to indicate tendency of a beneficial species to regularly occur in both orchard and non-orchard habitats. An unambiguous example of spillover is shown by *Anthocoris nemoralis* in southern France growing regions (Figure 13A). Resources available to natural enemies in one habitat allow population build-up and then “spillover” by these natural enemies into a second habitat, such as a pear orchard. Summaries of spillover are useful in allowing us to identify core species which—at least inferentially—disperse from non-orchard habitats into orchards and consequently should be targets for conservation via habitat manipulation. Less equivocally, the spillover graphic also identifies species which are unlikely to improve as sources of biocontrol in orchards even if native habitats and their psyllids were to be conserved. In southern France, for example, the predator *A. nemoralis* and two parasitoids, *T. insidiosus* and *P. mitratus*, are important components of psyllid biological control in orchards. Both *A. nemoralis* and *P. mitratus* also routinely attack non-pest psyllids (Figure 13A); thus, both species exhibit “spillover” between orchard and non-orchard habitats and both are likely to benefit reproductively from near-orchard presence of trees or shrubs which host psyllids. In contrast, *T. inisidiosus* has essentially an obligatory association with pear psyllids (Figure 13A) and only rarely has been collected from psyllid species other than pear psyllids. Conservation of psyllids in natural habitats would seem unlikely to strengthen biological control of pear psyllids by this parasitoid.

A similar graphic can be created for the pear-growing region in western North America (Figure 13B). As with *A. nemoralis* in Europe, the key true bug predators in Washington and Oregon routinely occur in native habitats, where they often co-occur with native psyllids on deciduous trees and shrubs as well as in pear orchards where they occur with *C. pyricola* (Figure 13B). Many of the native psyllids are in the family Psyllidae and genus *Cacopsylla* and consequently are related to the pear psyllid, *C. pyricola*. The parasitoid *T. insidiosus* in North America—as in Europe—seems to specialize on pear psyllids (in this case, *C. pyricola*). While it also parasitizes a willow psyllid (*Cacopsylla alba*; Table 2), there is some evidence that populations of the parasite in pears and in willow may constitute two non-mixing populations, as was discussed earlier. If true, there would be little spillover between orchard and non-orchard habitats by the parasitoid population which attacks pear psyllids (Figure 13B). Confirmation of this hypothesis would mean that native psyllids in near-orchard habitats likely have little effect on the biological control of *C. pyricola* by *T. insidiosus*.

Where the North American model departs from the European model is with behavior of the parasitoid *P. mitratus* (Figure 13B). This parasite is a visible component of the biological control community in European growing regions ([24]; see summary in [37]), where parasitism of pear psyllids may occasionally exceed 50% [15]. In contrast, rates of parasitism as reported from populations of pear psyllids in western North America rarely are higher than 1–2% and may often be lower than this rate ([109]; see also summary in [67]). As in Europe, *P. mitratus* in North America regularly parasitizes non-pest psyllids in the family Psyllidae (Table 2). North American hosts include a number of psyllids endemic to western North America, such as species on *Purshia* (*Purshivora* spp.), *Cercocarpus* (*Purshivora* spp.), *Ceanothus* (*Ceanothia* spp.), and *Salix* (*Cacopsylla* spp.). Several of these North American psyllids are multivoltine [29], and this life cycle would make the psyllids available to *P. mitratus* for a significantly larger part of the season than what is provided by the univoltine *Cacopsylla* targeted by the parasite in Europe (Figure 11 and Figure 12). It may be that availability of these multivoltine species in western North America has led to reduced spillover by *P. mitratus* into pear orchards (Figure 13B). Differences between European and North American populations in parasitism of pear psyllids could also indicate that the western Palearctic insect released into North America during the 1960s–1980s failed to establish. *Prionomitus mitratus* likely is naturally Holarctic [69], and it may be that the parasite currently present and established today in North America derived from the parasite which co-evolved in North America with the endemic Psyllidae fauna, rather than being derived from populations collected in Europe out of pear psyllid hosts and then released into North America.

### 7.2. Is Spillover Automatically a Sign of Movement?

Understanding insect movement is critical in implementing conservation biological control by manipulation of habitat diversity [110]. Logic would suggest that spillover between non-orchard and orchard habitats (Figure 13) is caused by movement between habitats. However, actual evidence that species are dispersing between habitats is difficult to obtain, and conclusions that species disperse between habitats generally are based upon inference or correlation rather than experimentation. One of the stronger cases that natural enemies arrive in pear orchards from native habitats is that for the parasitoid *P. mitratus* and its shift from univoltine “relay” hosts on hawthorn to the multivoltine *Cacopsylla pyri* in European pear orchards (described above in Figure 12). The switch to *C. pyri* may almost be obligatory, in that preference for hosts in the Psyllidae and *Cacopsylla* (Table 2) would force *P. mitratus* onto one of the few *Cacopsylla* in the study region that are multivoltine (i.e., *C. pyri*) once the univoltine hawthorn psyllids have disappeared (Figure 11 and Figure 12). Movement by *Anthocoris* predators from native habitats into pear orchards also has been examined, again indirectly through inference and correlation. Evidence includes observations that predator densities appear to be higher in orchards that neighbor stands of psyllid-infested willows [60] and correlative studies in which predator numbers were shown to climb in orchards at the same time that their numbers—and numbers of non-pest psyllids—began to decline in adjacent natural habitats [42,49].

Ultimately, confirming that natural enemies move between orchard and non-orchard habitats will require tracking of insects, possibly through use of a marker [110]. Application of external markers over enough of a geographic region to provide meaningful results may often be logistically difficult, although technological advances are producing solutions [111]. One shortcoming of external markers is that they may tell us about habitat sources of natural enemies but fail to identify the prey or host species used by the natural enemies in those habitats. An alternative approach would be the use of internal markers provided by the prey or host species, such as the protein or DNA of prey or hosts in the guts of the natural enemy. Basically, the predator or parasitoid itself would tell us its prey- or host-use history. One of the earliest (if not earliest) uses of molecular gut contents analysis to examine the diet of a predatory insect in fact was with *A. nemoralis* feeding on broom psyllids [85]. This study used immunological methods to detect the prey. This early approach has been overtaken by DNA-based methods using targeted or next-generation sequencing methods to examine dietary choices of predators as well as host use of parasitoids under field conditions [112,113]. Molecular assays of orchard-collected predators or parasitoids seemingly could be used to prove that DNA of non-orchard arthropods can be found in specimens collected from pear orchards. Psyllids which develop outside of the orchard would not be present in pear orchards as eggs or nymphs—due to their tendency towards monophagy; thus, the presence of DNA from psyllid species other than pear psyllids in a beneficial species would be strong supporting evidence that the beneficial had recently arrived from outside of the orchard. I am unaware of any use of this approach in orchards to identify alternative habitats and resources of dispersing natural enemies.

## 8. Concluding Remarks

While plant-diverse surroundings in crop-growing regions seem often to produce a more regionally diverse and abundant natural enemy fauna, effects on biological control in the crop is less easily quantified [12,13,14]. Tscharntke et al. 2007 [11] list several hypotheses to explain why there may be poor linkage between habitat diversity and conservation biological control, with many of these explanations focusing on physical or environmental properties of the landscape and crop and upon behavioral or other biological traits of the pests and their natural enemies. A common theme to these explanations is a need to understand the ecology and behavior of the natural enemies—including prey and host preferences or habitat preferences and their combined effects on population growth and movement—which in highly diverse systems can be a significant challenge. My objective in the current synthesis was to examine a core set of natural enemies and a single resource taxon (Psylloidea), to gain some understanding of whether key natural enemies of pear psyllids might benefit from the presence of native psyllids in near-orchard habitats.

This review shows that psyllids in native habitats are an important alternative resource for key natural enemies of pear psyllids in Europe and western North America. How valuable these alternative prey and hosts are to the beneficial species depends upon the natural enemy. Thus, while the true bug predators seem to be a regular component of the natural enemy community feeding on psyllids outside of orchards, the parasitoid *Trechnites insidiosus*—often a critical source of biological control in orchards—is rarely observed attacking psyllids other than the pear psyllids. The importance of native psyllids to natural enemies of pear psyllids is illustrated especially by the movement of natural enemies through the non-orchard landscape as they track the appearance and disappearance of their psyllid prey or hosts. This is shown most clearly by *Prionomitus mitratus* in Europe and by *Anthocoris* species both in Europe and North America. Untangling these patterns of behavior has required often intensive field research.

Conservation biological control through the manipulation of habitats was suggested as a possible tactic for pear psyllid control almost four decades ago [15]. A major obstacle in implementing this tactic is in understanding the movement of natural enemies—not just dispersal from native habitats into orchards (“spillover”) but also movements through the non-orchard landscape as the natural enemies track prey or host availability in these landscape habitats. Much can be learned merely through season-long monitoring of crop and non-crop habitats for the presence and absence of natural enemies, prey, and hosts. If the monitoring data were to be accompanied by molecular studies to identify prey DNA harbored by the predators, it should be possible to pinpoint which prey taxa in natural habitats deserve attention. Or, by collecting psyllid mummies and monitoring the emergence of parasitoids, it is possible to identify important alternative hosts for key parasitoids of pear psyllids as well as seasonal changes in the importance of different species. Lastly, I have suggested that collection of adult predators and parasitoids from pear orchards followed by assays to identify prey or host DNA harbored by specimens possibly could be used to identify psyllid species that had been fed upon or used as hosts by the predator or parasite specimen prior to its arrival in the pear orchard. As gut contents analyses become increasingly sophisticated, this strategy could prove to be useful in untangling some of these otherwise hidden interactions.

## Figures and Tables

**Figure 1 insects-15-00037-f001:**
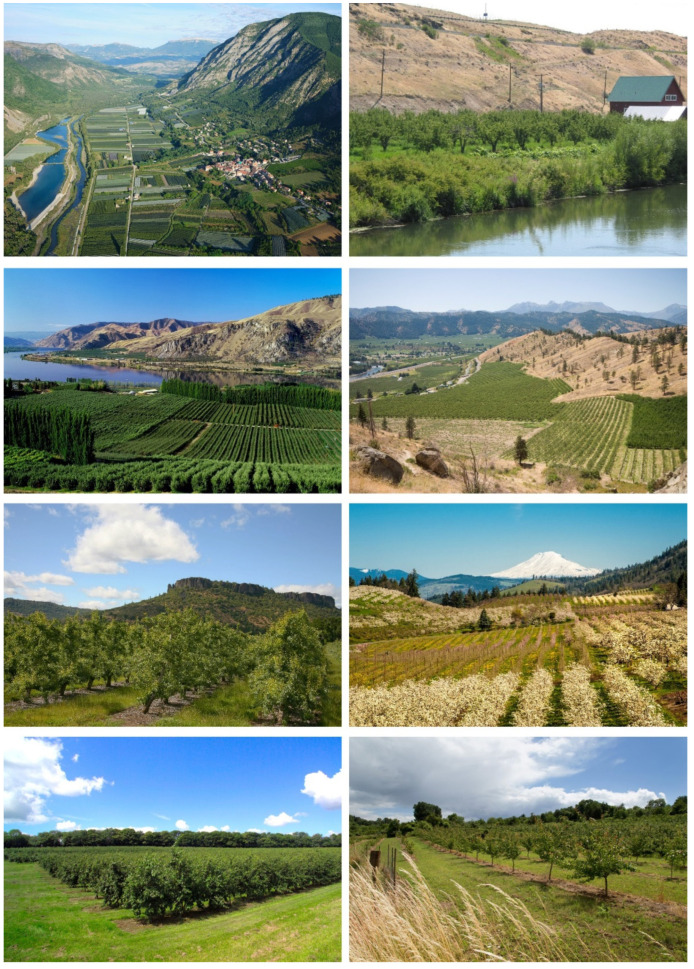
Examples of native riparian, deciduous woodland, and higher elevation shrubland abutting pear orchards in several geographic regions. Left column from top to bottom: (1) orchards adjacent to Durrance River in southeast France; (2) orchards and native habitat along Columbia River, Douglas County, northcentral Washington; (3) pear orchards with dryland shrub habitat in background, Medford Valley, Jackson County, southern Oregon; (4) pear orchard near Kent, England, with line of deciduous trees in background. Right column from top to bottom: (1) pear orchard adjacent to Yakima River, Yakima County, central Washington; (2) orchards in foothills east of Cascade Mountains, central Washington; (3) mixed deciduous woodland neighboring pear and apple orchards, Hood River County, northern Oregon (Mount Hood in background); (4) pear orchard with deciduous trees in background, Wacker, Germany. Photograph at top right is property of author. All other photographs licensed by and purchased through Alamy (https://www.alamy.com/, accessed on 25 September 2023).

**Figure 2 insects-15-00037-f002:**
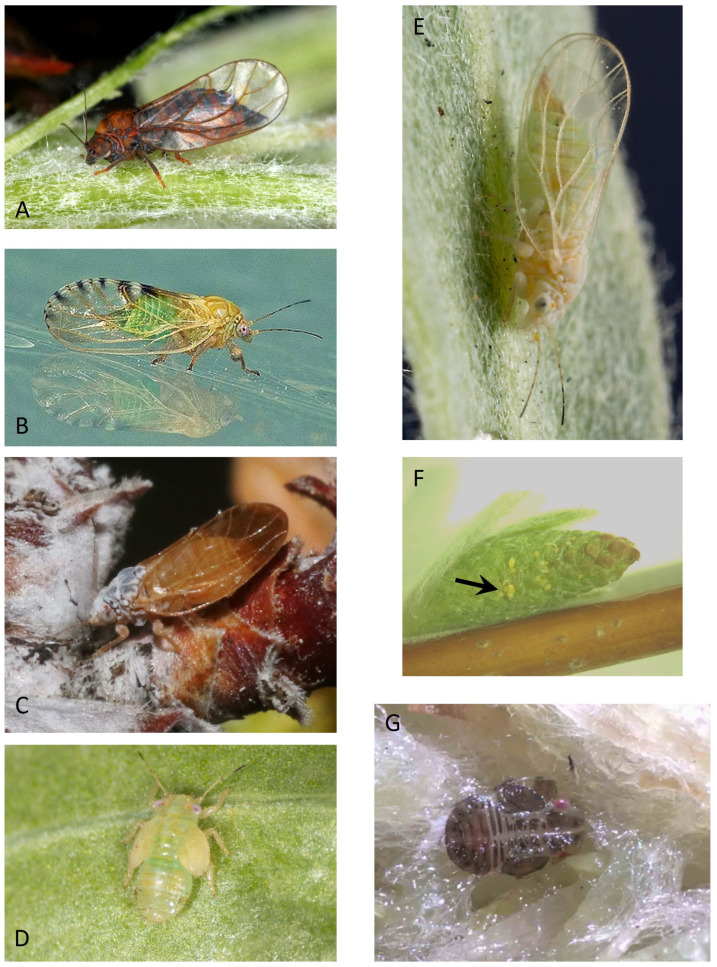
Photographs of a few psyllid species mentioned in text. (**A**) female *Cacopsylla pyrisuga* (Foerster), one of the few univoltine pear psyllids; photograph taken by Vladimir Motyčka (https://www.biolib.cz/en/image/id183554/, accessed on 5 June 2023) and used with permission; (**B**) female hawthorn psyllid *Cacopsylla crataegi* (Schrank); photograph taken by Ian Boyd (https://www.flickr.com, accessed on 26 September 2023) and available through a creative commons license; (**C**) female bitterbrush (*Purshia*) psyllid *Purshivora coryli* (Patch); photograph taken by James Bailey (https://bugguide.net/node/view/1526238, accessed on 17 November 2023) and used with permission; (**D**) late-instar nymph of *Cacopsylla ribesiae* (Crawford); (**E**) female of the willow psyllid *Cacopsylla alba* (Crawford) on its *Salix exigua* host; photograph courtesy of Alice Abela (https://bugguide.net/node/view/1231999, accessed on 17 November 2023) and used with permission; (**F**) eggs (arrow) of *C. alba* inserted into young catkin of *S. exigua*; (**G**) late-instar nymph of a catkin-specialized psyllid, *Cacopsylla* sp., in “fluff” of a mature catkin on *Salix prolixa*. Photographs (**D**,**F**,**G**) are property of the author.

**Figure 3 insects-15-00037-f003:**
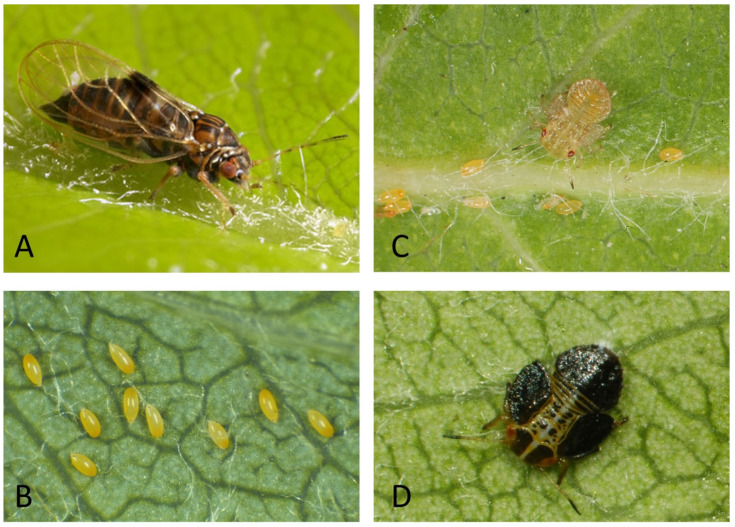
Life stages of psyllids, showing stages of the pear psyllid *Cacopsylla pyricola* (Foerster) as an example. (**A**). adult female; photograph of Brian Valentine and used with permission (https://www.britishbugs.org.uk/homoptera/Psylloidea/Psylla_pyricola.html, accessed on 29 November 2023); (**B**–**D**). Eggs, mid-instar nymph, and last-instar nymph; photographs provided by Elizabeth Beers, Washington State University, and used with permission.

**Figure 4 insects-15-00037-f004:**
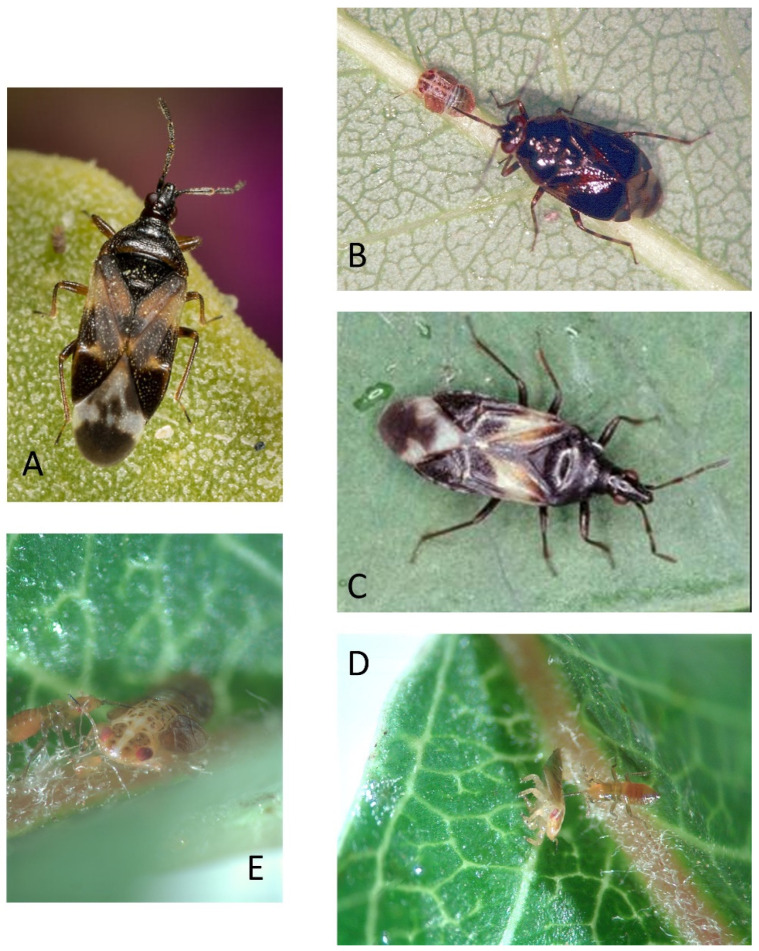
Key predators targeted by this review. (**A**) *Anthocoris nemoralis*; photograph taken by Alice Abela (https://bugguide.net/node/view/1060544, accessed on 17 November 2023) and used with permission; (**B**) *Deraeocoris brevis* feeding on late-instar nymph of the pear psyllid *Cacopsylla pyricola* (Washington State); (**C**) *Anthocoris antevolens* (Washington State); (**D**,**E**) nymphal *Anthocoris antevolens* feeding on *C. pyricola* nymph (Washington State). Photographs (**B**–**E**) are property of author.

**Figure 5 insects-15-00037-f005:**
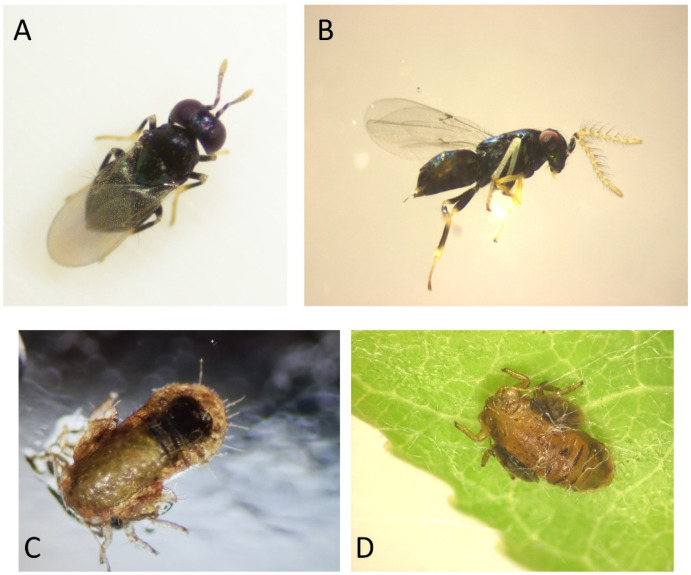
Key parasitoids targeted by this review. (**A**) adult *Trechnites insidiosus* (female); (**B**) adult *Prionomitus mitratus* (male); (**C**) late-instar nymph (mummy) of *C. pyricola* parasitized by *T. insidiosus*; (**D**) late-instar nymph (mummy) of a willow psyllid (*Cacopsylla* sp.) parasitized by *P. mitratus*. Specimens from central Washington State. (**A**,**C**) provided by Rebecca Schmidt-Jeffris, USDA-ARS, and used with permission. (**B**,**D**) are property of the author.

**Figure 6 insects-15-00037-f006:**
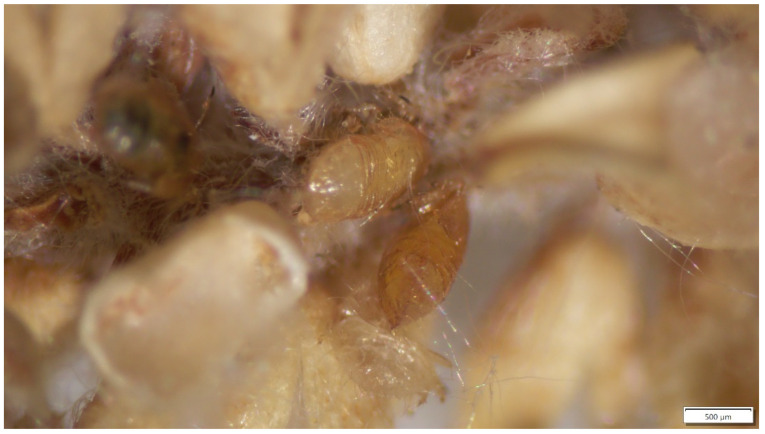
Two mummies of the willow psyllid *Cacopsylla alba* parasitized by *T. insidiosus* (central Washington State); mummies buried in catkin of *Salix exigua* [29]. Photograph provided by Rebecca Schmidt-Jeffris, USDA-ARS, and used with permission.

**Figure 7 insects-15-00037-f007:**
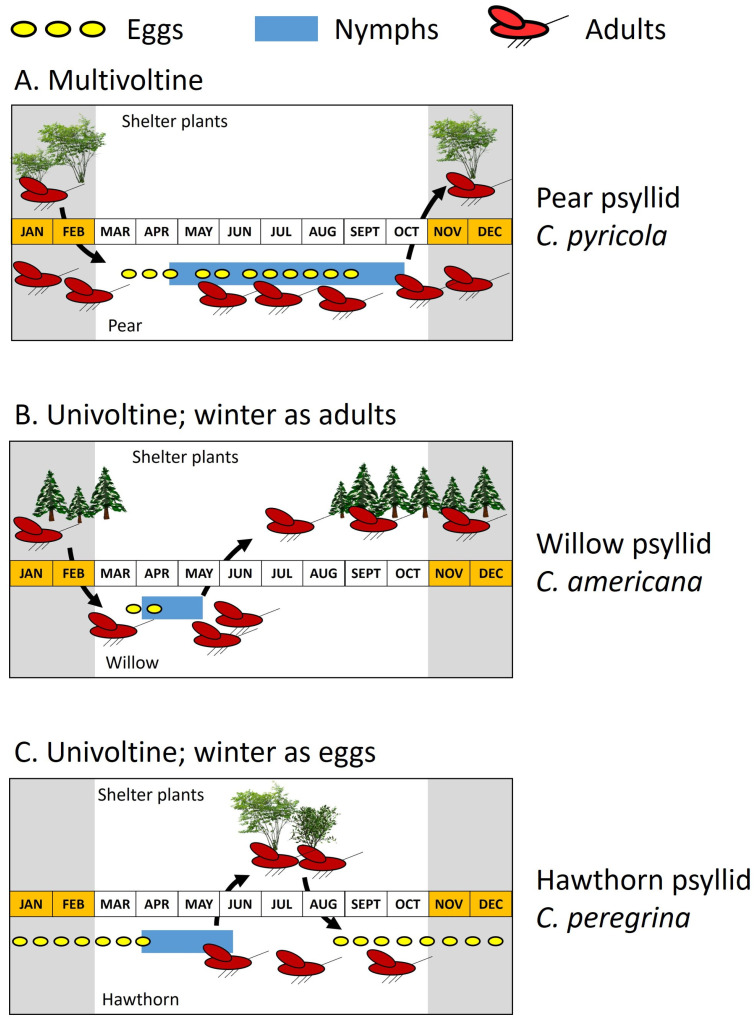
Contrasts in life cycles of *Cacopsylla*. Lower half of each graphic (below the month timeline) shows psyllid presence on host plant; upper half shows adult presence on shelter plants. Yellow ovals: egg presence; blue bars: nymph presence. Arrows show approximate timing of adult movement between host and shelter plants. (**A**) Phenology of the pear psyllid *C. pyricola* (central Washington State); (**B**) phenology of a catkin-specialized *Cacopsylla* species on willow in central Washington State; (**C**) phenology of the hawthorn psyllid *C. peregrina* (central Europe). Phenological data are from [29,100].

**Figure 8 insects-15-00037-f008:**
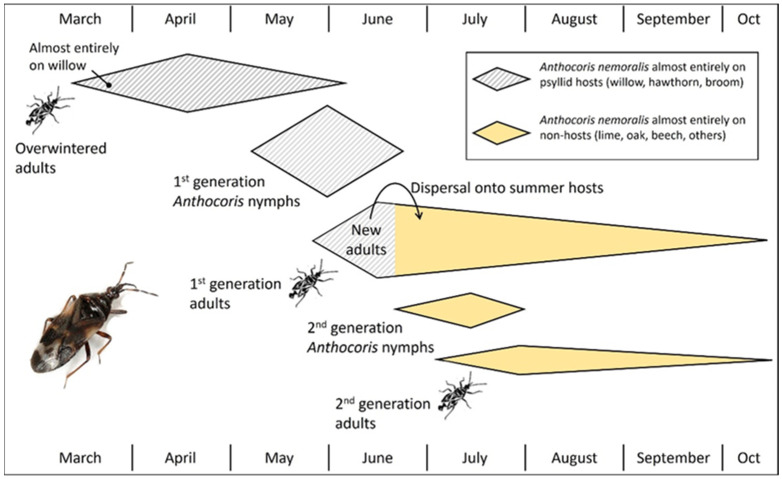
Somewhat idealized figure describing the seasonally “nomadic” behavior of a population of *Anthocoris nemoralis* in southern England. Graphic modified from Figure 11 in Anderson 1962b [19].

**Figure 9 insects-15-00037-f009:**
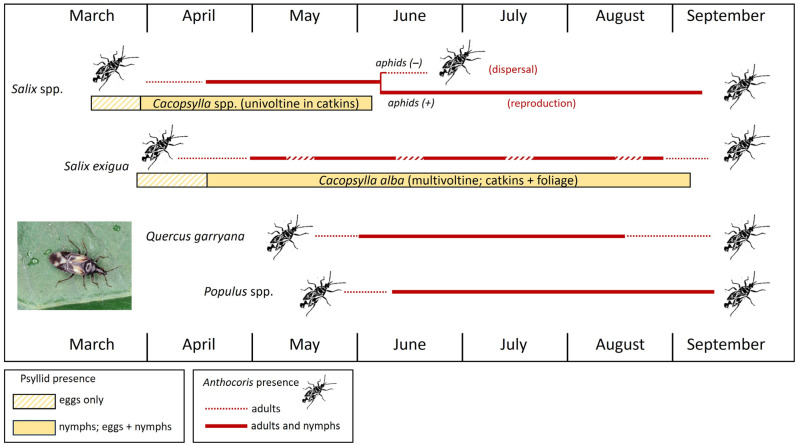
Somewhat idealized figure describing the seasonally “nomadic” behavior of *Anthocoris tomentosus* and *A. antevolens* in a riparian/deciduous woodland habitat of central Washington State (Yakima County). *Populus* spp.: *Populus trichocarpa*, *Populus nigra*, and *Populus tremuloides*. The graphic for “*Salix* spp.” shows that the predators fail to disperse following disappearance of the univoltine willow psyllids if the willow becomes infested by aphids but do disperse if aphids fail to appear. Interruptions shown in the horizontal bar for *Salix exigua* depict the spatially patchy distribution of the predators on sandbar willow as they track presence of catkins and psyllid infestations. Graphic produced from data in Horton and Lewis 2000 [25] and unpublished collecting records [29].

**Figure 10 insects-15-00037-f010:**
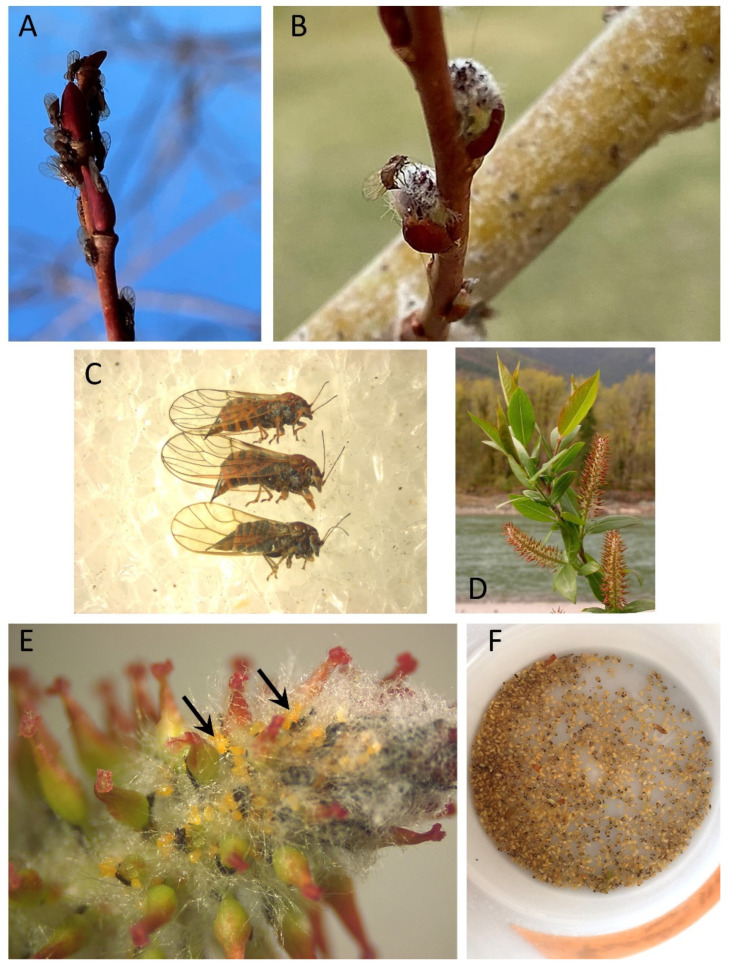
Photographs showing phenology from late winter through spring and population build-up of univoltine *Cacopsylla* species in catkins of *Salix* spp. (central Washington State). (**A**) Overwintered adults accumulating on unopened buds of *Salix prolixa* following arrival from coniferous shelter plants (6 March 2022); (**B**) female *Cacopsylla* ovipositing into newly opened bud of *Salix* sp. (21 March 2022); (**C**) mix of *Cacopsylla* species (overwintered females) from *Salix prolixa*, illustrating the taxonomic complexity of psyllids within this group; (**D**) mature female catkins of *S. prolixa*; (**E**) eggs and first-instar nymphs (arrows) of *Cacopsylla* spp. in female catkin of *Salix prolixa* (28 March 2017); (**F**) “soup” of late-instar nymphs of *Cacopsylla* spp. extracted into preservative via a Berlese funnel, from ~50 female catkins of *Salix prolixa* (7 May 2017). All photographs are property of the author.

**Figure 11 insects-15-00037-f011:**
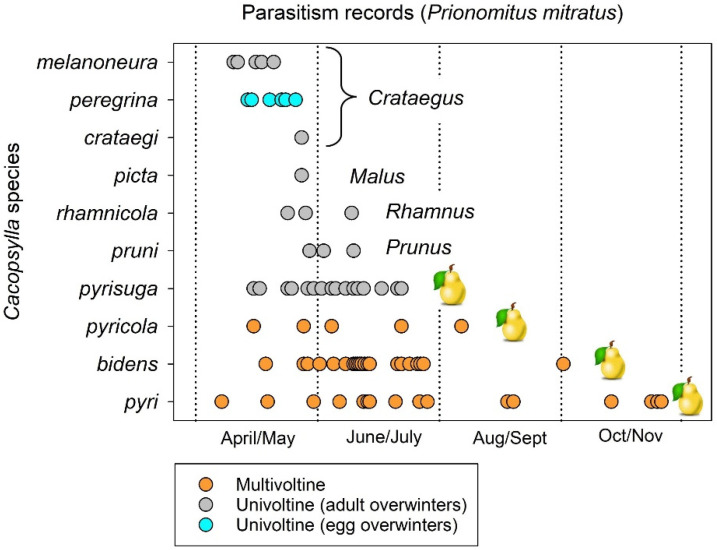
Seasonal timetable of emergence of the parasitoid *Prionomitus mitratus* from mummies of different species of univoltine and multivoltine *Cacopsylla*. Mummies were collected over several years and at multiple locations in Serbia and monitored for emergence of the parasitoid. From data in Jerinić-Prodanović et al. 2010 and 2019 [28,107].

**Figure 12 insects-15-00037-f012:**
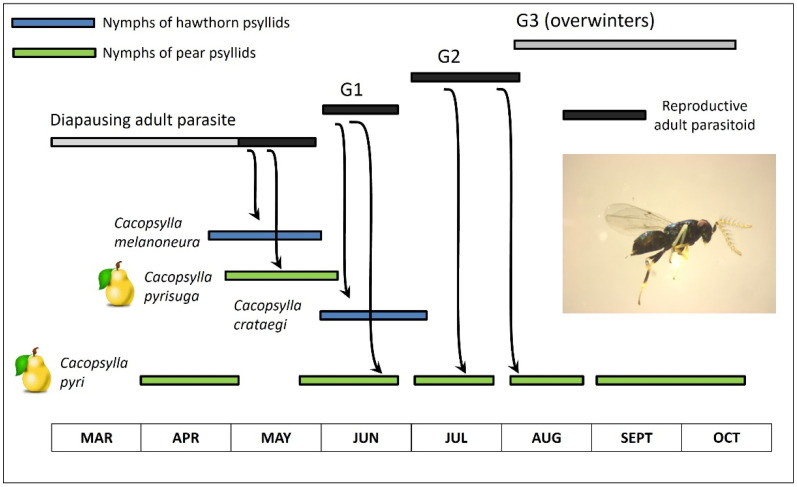
Sequence in use of univoltine and multivoltine *Cacopsylla* hosts by the parasitoid *Prionomitus mitratus* in a pear-growing region of southern France. Blue bars show seasonal availability of univoltine hawthorn psyllids, while green bars show seasonal availability of univoltine (*C. pyrisuga*) and multivoltine (*C. pyri*) pear psyllids. “G” indicates generation number of the parasitoid. Modified from Figure 2 of Nguyen and Delvare 1982 [23]. Another version of this graphic is in Civolani et al. 2023 [1].

**Figure 13 insects-15-00037-f013:**
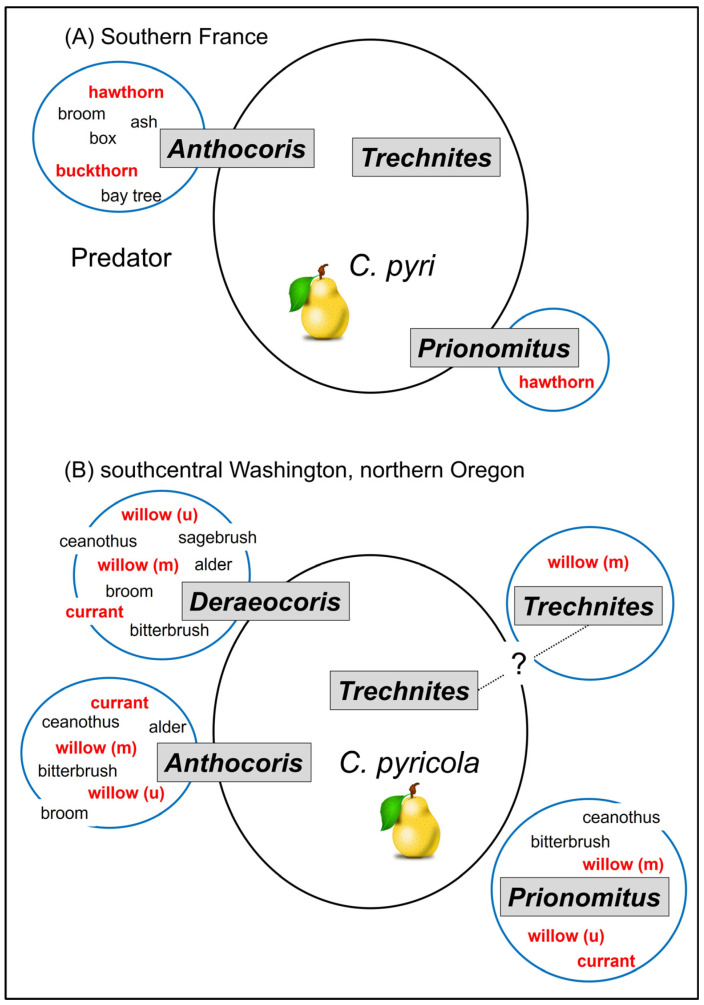
“Spillover” by key natural enemies of pear psyllids between native habitats and pear orchards. Spillover is defined as regular presence of the natural enemy in native habitats on plant taxa which host psyllids (blue ovals in figures) and in pear orchards (larger, black ovals in figures). Taxa which connect both to natural habitats and pear orchards (e.g., *Anthocoris* spp.) can be said to exhibit meaningful spillover between habitats. (**A**): southern France (from data in [15,17,22,24,41]); (**B**): southcentral Washington and northern Oregon (from data in [25,26,63] and personal collecting of beneficial species [29]). Plant species in red font are hosts to *Cacopsylla* psyllids; (m)—multivoltine willow psyllid (*C. alba*); (u)—assemblage of catkin-specialized and univoltine *Cacopsylla* spp. on willows.

**Table 1 insects-15-00037-t001:** Psyllid species shown or inferred to be preyed upon by *Anthocoris nemoralis*, *Anthocoris tomentosus*, *Anthocoris antevolens*, and *Deraeocoris brevis* and host plant of psyllid. Supporting evidence categorized according to strength of the evidence: “Direct evidence” or “Other evidence” (see text). Records summarized from multiple sources, including personal observations or collecting [15,17,19,21,24,25,27,29,42,46,51,57,64,82,85,86,87,88,89,90,91,92,93,94,95,96,97,98].

Predator Species	Supporting Evidence	Psyllid Species	Psyllid Family	Psyllid Host Plant
*Anthocoris nemoralis*	Direct evidence	*Cacopsylla bidens* (Šulc), *Cacopsylla notata* (Flor), *Cacopsylla pyri* (L.), *Cacopsylla pyricola* (Foerster), *Cacopsylla pyrisuga* (Foerster)	Psyllidae	Pear (*Pyrus communis*; Rosaceae)
		*Cacopsylla melanoneura* (Foerster), *Cacopsylla peregrina* (Foerster)	Psyllidae	European hawthorn (*Crataegus monogyna*; Rosaceae)
		*Cacopsylla visci* (Curtis)	Psyllidae	Mistletoes (*Loranthus europaeus*; Loranthaceae; *Viscum album*; Santalaceae)
		*Cacopsylla myrthi* (Puton)	Psyllidae	Buckthorn (*Rhamnus* sp. Rhamnaceae)
		*Cacopsylla mali* (Schmidberger)	Psyllidae	Apple (*Malus domestica*; Rosaceae)
		*Cacopsylla* spp. (assemblage of species)	Psyllidae	Willow (*Salix* spp.; Salicaceae)
		* *Acizzia uncatoides* (Ferris & Klyver)	Psyllidae	Acacia (*Acacia longifolia*; Fabaceae)
		*Arytaina genistae* (Latreille), *Arytainilla spartiophila* (Foerster)	Psyllidae	Scotch broom (*Cytisus scoparius*; Fabaceae)
		*Camarotoscena speciosa* (Flor)	Liviidae	Black poplar (*Populus nigra*; Salicaceae)
		*Psyllopsis discrepans* (Flor), *Psyllopsis fraxini* (L.), *Psyllopsis machinosus* Loginova, *Psyllopsis meliphila* Löw, *Psyllopsis repens* Loginova	Liviidae	Ash (*Fraxinus* spp.; Oleaceae)
		*Trioza urticae* (L.)	Triozidae	Stinging nettle (*Urtica dioica*; Urticaceae)
		* *Trioza adventicia* Tuthill	Triozidae	Roseapple (*Syzygium* sp.; Myrtaceae)
		*Lauritroza alacris* (Flor)	Triozidae	Bay laurel (*Laurus nobilis*; Lauraceae)
		*Agonoscena cisti* (Puton), *Agonoscena targionii* (Lichenstein)	Aphalaridae	Mastic (*Pistacia lentiscus*; Anacardiaceae)
		* *Macrohomotoma gladiata* Kuwayama	Carsidaridae	Curtain fig (*Ficus microcarpa*; Moraceae)
		* *Calophya schini* Tuthill	Calophyidae	Peppertree (*Schinus molle*; Anacardiaceae)
	Other evidence	*Cacopsylla crataegi* (Schrank), *Cacopsylla affinis* (Löw)	Psyllidae	European hawthorn
		*Cacopsylla alaterni* (Foerster), *Cacopsylla rhamnicola* (Scott)	Psyllidae	Buckthorn (*Rhamnus alaternus*; Rhamnaceae)
		*Cacopsylla ambigua* (Foerster), *Cacopsylla saliceti* (Foerster)	Psyllidae	Willow (*Salix alba*; Salicaceae)
		*Cacopsylla hippophaes* (Foerster)	Psyllidae	Seaberry (*Hippophae rhamnoides*; Elaeagnaceae)
		*Cacopsylla pulchella* (Löw)	Psyllidae	Redbud (*Cercis* sp.; Fabaceae)
		* *Cacopsylla fulguralis* (Kuwayama)	Psyllidae	Silverberry (*Elaeagnus* spp.; Elaeagnaceae)
		* *Acizzia jamatonica* (Kuwayama)	Psyllidae	Silk tree (*Albizia julibrissin*; Fabaceae)
		*Spanioneura buxi* (L.), *Spanioneura fonscolombii* Foerster	Psyllidae	Common box (*Buxus sempervirens*; Buxaceae)
		*Psylla alni* (L.), *Psylla foersteri* (Flor)	Psyllidae	Alder (*Alnus* spp.; Betulaceae)
		* *Platycorypha nigrivirga* Burckhardt	Psyllidae	Rosewood (*Tipuana tipu*; Fabaceae)
		*Psyllopsis fraxinicola* (Foerster)	Liviidae	Ash
		*Trioza marginepunctata* Flor, *Trioza rhamni* (Schrank)	Triozidae	Buckthorn (*Rhamnus alaternus*)
		* *Homotoma ficus* (L.)	Carsidaridae	Common fig (*Ficus carica*; Moraceae)
		* *Glycaspis brimblecombei* Moore	Aphalaridae	Eucalyptus (*Eucalyptus* spp.; Myrtaceae)
*Anthocoris tomentosus* and *Anthocoris antevolens*	Direct evidence	* *Cacopsylla pyricola*	Psyllidae	Pear
		*Cacopsylla* spp. (assemblage of univoltine species generally associated with catkins)	Psyllidae	Willows (*Salix prolixa*, *S. scouleriana*, *Salix* spp.; Salicaceae)
		*Cacopsylla alba* (Crawford) (multivoltine; catkin + foliar feeder)	Psyllidae	Sandbar willow (*Salix exigua*; Salicacaeae)
	Other evidence	*Cacopsylla pararibesiae* (Jensen), *Cacopsylla ribesiae* (Crawford)	Psyllidae	Golden currant (*Ribes aureum*; Grossulariaceae)
		*Cacopsylla* sp.	Psyllidae	Buffaloberry (*Shepherdia canadensis*; Elaeagnaceae)
		* *Cacopsylla peregrina*	Psyllidae	European hawthorn
		*Purshivora* spp. (assemblage of species)	Psyllidae	Antelope bitterbrush (*Purshia tridentata*; Rosaceae)
		*Ceanothia* sp.	Psyllidae	Ceanothus (*Ceanothus* spp.; Rhamnaceae)
		*Nyctiphalerus vermiculosus* (Crawford)	Psyllidae	Ceanothus
		*Psylla floccosa* Patch	Psyllidae	Alder (*Alnus* spp.; Betulaceae)
		* *Arytaina genistae*, * *Arytainilla spartiophila*	Psyllidae	Scotch broom
*Deraeocoris brevis*	Direct evidence	* *Cacopsylla pyricola*	Psyllidae	Pear
		*Cacopsylla alba*	Psyllidae	Sandbar willow
	Other evidence	*Cacopsylla* spp. (assemblage of univoltine species)	Psyllidae	Willows
		*Purshivora* spp. (assemblage of species)	Psyllidae	Antelope bitterbrush
		*Ceanothia* sp.	Psyllidae	Ceanothus
		*Nyctiphalerus vermiculosus*	Psyllidae	Ceanothus
		*Psylla floccosa*	Psyllidae	Alder
		* *Arytaina genistae*, * *Arytainilla spartiophila*	Psyllidae	Scotch broom
		*Craspedolepta* spp. (assemblage of species)	Aphalaridae	Sagebrush (*Artemisia tridentata*; Asteraceae)
		*Trioza albifrons* Crawford	Triozidae	Stinging nettle (*Urtica gracilis*; Urticaceae)
		*Bactericera cockerelli* (Šulc)	Triozidae	Bittersweet nightshade (*Solanum dulcamara*); matrimony vine (*Lycium barbarum*) (Solanaceae)

* Psyllid species is from outside of the native range of the predator.

**Table 2 insects-15-00037-t002:** Host records for *Trechnites insidiosus* and *Prionomitus mitratus* from the western Palearctic region and North America and host plant of psyllid. Records were summarized from multiple sources, including personal observations or collecting [28,29,69,80,83].

Parasitoid Species	Geographic Region	Parasitoid Host Species	Psyllid Family	Psyllid Host Plant
*Trechnites insidiosus*	Western Palearctic	*Cacopsylla bidens* *, *Cacopsylla pyri*, *Cacopsylla pyrisuga*, *Cacopsylla pyricola*	Psyllidae	Pear (*Pyrus communis*; Rosaceae)
		*Cacopsylla crataegi*	Psyllidae	European hawthorn (*Crataegus monogyna*; Rosaceae)
		*Cacopsylla breviantennata* (Flor)	Psyllidae	Mountain ash (*Sorbus aria*; Rosaceae)
	North America	*Cacopsylla pyricola*	Psyllidae	Pear
		*Cacopsylla alba*	Psyllidae	Sandbar willow (*Salix exigua*; Salicaceae)
*Prionomitus mitratus*	Western Palearctic	*Cacopsylla bidens*, *Cacopsylla pyri*, *Cacopsylla pyrisuga*, *Cacopsylla pyricola*	Psyllidae	Pear
		*Cacopsylla peregrina*, *Cacopsylla melanoneura*, *Cacopsylla crataegi*	Psyllidae	European hawthorn
		*Cacopsylla mali*, *Cacopsylla picta* (Foerster)	Psyllidae	Apple (*Malus domestica*; Rosaceae)
		*Cacopsylla pruni* (Scopoli)	Psyllidae	Plum; Sloe (*Prunus* spp.; Rosaceae)
		*Cacopsylla rhamnicola*	Psyllidae	Buckthorn (*Rhamnus* spp.; Rhamnaceae)
		*Acizzia jamatonica*	Psyllidae	Silk tree (*Albizia julibrissin*; Fabaceae)
		*Psylla foersteri* (Flor)	Psyllidae	Black alder (*Alnus glutinosa*; Betulaceae)
		*Livilla retamae* (Puton)	Psyllidae	Broom (*Retama* sp.; Fabaceae)
		*Agonoscena pistaciae* Burckhardt & Lauterer	Aphalaridae	Pistachio (*Pistacia* sp; Anacardiaceae)
		*Macrohomotoma gladiata*	Carsidaridae	Curtain fig (*Ficus microcarpa*; Moraceae)
	North America	*Cacopsylla pyricola*	Psyllidae	Pear
		*Cacopsylla alba*	Psyllidae	Sandbar willow (*Salix exigua*; Salicaceae)
		*Cacopsylla* spp. (assemblage of species)	Psyllidae	Willows (*Salix prolixa*, *Salix* spp.; Salicaceae)
		*Cacopsylla ribesiae*	Psyllidae	Currant (*Ribes* sp.; Grossulariaceae)
		*Purshivora* spp. (assemblage of species)	Psyllidae	Antelope bitterbrush (*Purshia tridentata*; Rosaceae)
		*Purshivora media* (Tuthill)	Psyllidae	Mountain mahogany (*Cercocarpus* spp.; Rosaceae)
		*Pexopsylla cercocarpi* Jensen	Psyllidae	Mountain mahogany
		*Psylla alni* (L.), *Psylla floccosa*	Psyllidae	Alder (*Alnus* spp.; Betulaceae)
		*Ceanothia ceanothi* (Crawford), *Ceanothia essigi* (Jensen), *Ceanothia insolita* (Tuthill), *Ceanothia fuscipennis* (Crawford), *Ceanothia minuta* (Crawford), *Ceanothia robusta* (Crawford)	Psyllidae	Ceanothus (*Ceanothus* spp.; Rhamnaceae)
		*Nyctiphalerus vermiculosus* (Crawford)	Psyllidae	Ceanothus

* Referred to as *Cacopsylla vasiljevi* in some records.

## Data Availability

No new data were created or analyzed in this study. Data sharing is not applicable to this article.

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
