# Peer review of "Psyllids in Natural Habitats as Alternative Resources for Key Natural Enemies of the Pear Psyllids (Hemiptera: Psylloidea)"

_insects, 2024, doi:10.3390/insects15010037_

Round 1

Reviewer 1 Report

Comments and Suggestions for Authors

This is article is a literature on alternative prey and the host plants of alternative prey for predators of pear psyllids. I found the review very enthusing. I think this will be a useful reference for researchers of pear IPM,  it outlines some excellent suggestions for new research, and could be useful to other professionals interested in benefits of native plants. The use of photos and figures is excellent and very helpful, contributing to the strength of this resource. I have a few comments that are merely suggestions.

L55 suggest replace "mould" with "mold". A mould is something that holds a malleable object.

L150. Instead of stating that the list is below. The list of 6 species could be presented in place of the current sentence. It is not until 3 paragraphs later that the list appears, so it seems like quite a tease the way it is currently worded.

Table 1. I think clarification is needed whether all of these observations come from North America or is this a world review? The footnote is not clear because I don't think it is very clearly defined in the main text (or it was easy enough for me to miss) what is the native ranges of all the species described. The addition of a geographic range column like in table 2 may clarify.

It may be helpful to cite any recommendable literature (if there are any) that researchers could use to assist in identification of the relevant species. Since the relative importance of pear psyllid predators may vary between regions and differ from those in this review, it would be useful for people across regions to gain confidence in whether what they have overlaps with the information here.

Author Response

L55 suggest replace "mould" with "mold". A mould is something that holds a malleable object. I have made this change (line 55 in revision).

L150. Instead of stating that the list is below. The list of 6 species could be presented in place of the current sentence. It is not until 3 paragraphs later that the list appears, so it seems like quite a tease the way it is currently worded. I have added information on the 6 taxa at the requested location (lines 152-153).

Table 1. I think clarification is needed whether all of these observations come from North America or is this a world review? The footnote is not clear because I don't think it is very clearly defined in the main text (or it was easy enough for me to miss) what is the native ranges of all the species described. The addition of a geographic range column like in table 2 may clarify. I have added text in lines 353-355 indicating that the records for the European Anthocoris nemoralis do indeed include records from North America, and that the predator is established there (in California). I did not revise the table.

It may be helpful to cite any recommendable literature (if there are any) that researchers could use to assist in identification of the relevant species. Since the relative importance of pear psyllid predators may vary between regions and differ from those in this review, it would be useful for people across regions to gain confidence in whether what they have overlaps with the information here. I added a paragraph summarizing where one would find this information (lines 297-305). These new literature citations were added to the References Cited (lines 984-991). This was a good suggestion.

Reviewer 2 Report

Comments and Suggestions for Authors

The manuscript “Psyllids in natural habitats as alternative resources for key natural enemies of the pear psyllids (Hemiptera: Psylloidea)” analyses the role of different psyllids inside and outside pear orchards as alternative preys/hosts for natural enemies of pear psyllids. In particular, their suitability based on their closeness to pear psyllids and their seasonal life cycles is examined. The considerations emerging from this review are very interesting and highlight the importance of preserving natural environments with a view to conservative biological control of pear tree psyllids. The text is well organized and clearly written.

I have just some minor remarks:

-          Introduction: speaking of damage caused by pear psyllids, no reference is made to the transmission of the phytoplasma causing Pear Decline, a disease with a severe impact on pear cultivation. I suggest that the author add some information on this subject with appropriate bibliography.

-          Caption of Fig. 4 and 5: Scientific names should be in Italics

-          Table 1: try to better align the psyllid species, otherwise it would be confusing

-          Figure 7: In my opinion the two charts for fig 7b are redundant: choose only one of them, the one from hawthorn or the one from willow, as an example

-          Figure 7, picture of C. melanoneura: I have some doubts that it is really C. melanoneura. Or maybe the photo has been retouched in its proportions and that is why the psyllid looks more elongated than usual. Please check it.

-          Pag. 28, Psyllids which develop outside of the orchard would not be present in pear orchards…..: Please consider that there are several psyllids that can occasionally feed on pear trees such as Cacopsylla melanoneura or Cacopsylla mali (See for instance  Lauterer 1999, Acta Musei Moraviae, Scientiae biologicae, 84: 71-151)

Author Response

Introduction: speaking of damage caused by pear psyllids, no reference is made to the transmission of the phytoplasma causing Pear Decline, a disease with a severe impact on pear cultivation. I suggest that the author add some information on this subject with appropriate bibliography. I have added this text (lines 52-53)

-          Caption of Fig. 4 and 5: Scientific names should be in Italics. I have corrected this mistake.

-          Table 1: try to better align the psyllid species, otherwise it would be confusing. I have reformatted both tables so that the psyllid species align better.

-          Figure 7: In my opinion the two charts for fig 7b are redundant: choose only one of them, the one from hawthorn or the one from willow, as an example. I have eliminated the hawthorn example, and kept the willow example, thus eliminating the redundancy.

-          Figure 7, picture of C. melanoneura: I have some doubts that it is really C. melanoneura. Or maybe the photo has been retouched in its proportions and that is why the psyllid looks more elongated than usual. Please check it. The photograph of C. melanoneura has been removed completely from the manuscript (I was unable to receive permission to use it).

-          Pag. 28, Psyllids which develop outside of the orchard would not be present in pear orchards…..: Please consider that there are several psyllids that can occasionally feed on pear trees such as Cacopsylla melanoneura or Cacopsylla mali (See for instance  Lauterer 1999, Acta Musei Moraviae, Scientiae biologicae, 84: 71-151). I added some text (lines 771-773) clarifying the original statement, and now state that I am referring only to the egg and nymphal stages. 

Round 2

Reviewer 2 Report

Comments and Suggestions for Authors

The author has adequately fulfilled all proposed revisions